# Factors causing emergency medical care overload during heatwaves: A Delphi study

**Matteo Paganini** [1,2,3]*, **Hamdi Lamine** [2,4], **Francesco Della Corte**[2,3], **Ives Hubloue**[5], **Luca Ragazzoni**[2,4], **Francesco Barone-Adesi**[2,3]

1 Department of Biomedical Sciences, University of Padova, Padova, Italy, 2 CRIMEDIM—Center for Research and Training in Disaster Medicine, Humanitarian Aid and Global Health, Università del Piemonte Orientale, Novara, Italy, 3 Department of Translational Medicine, Università del Piemonte Orientale, Novara, Italy, 4 Department for Sustainable Development and Ecological Transition, Università del Piemonte Orientale, Vercelli, Italy, 5 Research Group on Emergency and Disaster Medicine, Vrije Universiteit Brussel, Brussels, Belgium

* matteo.paganini@unipd.it

**Data Availability Statement:** All relevant data are within the paper and its Supporting Information files.

## Abstract

Heatwaves pose an important risk for population health and are associated with an increased demand for emergency care. To find factors causing such overload, an online Delphi study included 15 experts in emergency medicine, disaster medicine, or public health. One open-ended question was delivered in the first round. After content analysis, the obtained statements were sent to the experts in two rounds to be rated on a 7-point linear scale. Consensus was defined as a standard deviation ≤ 1.0. Thirty-one statements were obtained after content analysis. The experts agreed on 18 statements, mostly focusing on the input section of patient processing and identifying stakeholders, the population, and primary care as targets of potential interventions. Additional dedicated resources and bed capacity were deemed important as per throughput and output sections, respectively. These findings could be used in the future to implement and test solutions to increase emergency healthcare resilience during heatwaves and reduce disaster risk due to climatic change.

## Introduction

Excess greenhouse emissions related to human activities are causing a constant increment in global mean temperature. Consequences include climate change and increased extreme weather events, with devastating effects on communities and ecosystems [1]. Although no unique definition of heatwaves has been established, they are generally described as prolonged periods of unusual heat whose frequency and intensity are increasing [2]. In addition, a growing evidence indicates an association between heatwaves and populations' health, particularly with the mortality and morbidity of vulnerable strata of the exposed communities [3–5]. However, several knowledge gaps still have to be investigated to improve preparedness and response to heatwaves, such as the strain on healthcare systems [3,4]. Some retrospective studies found an association between heatwaves and the increase in health services demand across different countries, specifically in emergency medical care [6–8]. However, it is not clear

**Funding:** The authors received no specific funding for this work.

**Competing interests:** The authors have declared that no competing interests exist.

whether this phenomenon impacts systems' efficiency or patient flow in emergency departments (EDs) or emergency medical services (EMSs) [6]. In the era of emergency medical care crisis and ED overcrowding [9], there is a need to identify the factors causing emergency medical care overload and provide stakeholders with valuable information to improve EMSs and EDs preparedness to face heatwaves-related surges of patients.

This study aimed to find the factors causing an overload of emergency medical care–both pre-hospital (EMSs) or in-hospital (EDs)–during heatwaves, pursuing United Nations' Sustainable Development Goal No. 13th: Climate Action, and specifically aiming to "*strengthen resilience and adaptive capacity to climate-related hazards and natural disasters in all countries*".

## Materials and methods

This study was an expert opinion study based on the Delphi methodology, modified to be delivered and conducted online [10].

### Participants

After obtaining formal authorization from the institutional ethics committee (No. 160–18/02/2022), the experts' recruiting phase started on July 20th, 2022. To be included, participants' expertise had to fall in one of the following fields: emergency medicine; disaster medicine; or public health. The number of experts was set to a minimum of 12, according to Delphi methodology literature [10]. Anonymity among panel members was ensured throughout the study; only two researchers handling recruitment and analysis accessed personal details.

Authors of literature dealing with heatwaves and overload of healthcare or emergency medical care systems, retrieved through recent reviews [4,6], were contacted via email. Two reminder emails were sent every 3 weeks if unresponsive. To ensure sufficient experts to conclude the study after possible drop-outs, a last recruitment round was held during the Disaster Medicine section's meeting at the European Society for Emergency Medicine (EUSEM) congress in October 2022. Recruitment was ended on October 31st, 2022.

### Structure

The modified Delphi was structured in three rounds. After completing a review of the literature [4,6], the authors convened that the retrieved literature was insufficient to elaborate robust statements for the first round. Therefore, the following open-ended question was created to collect five to ten experts' answers: "What are the factors causing emergency medical care overload during heatwaves?". To ensure content and face validity, this question was pilot-tested with three Ph.D. students (with expertise in the field) and reviewed for accuracy by three senior faculty members from the Research Center in Emergency and Disaster Medicine (CRIMEDIM), Università del Piemonte Orientale, Novara, Italy. Age, country, and field of expertise were also tracked. Content analysis was then performed by one author and blindly reviewed by another author to ensure its validity. In the second round, experts were asked to rate each statement for importance on a 7-point linear scale, with 1 meaning "Not at All Important" and 7 meaning "Extremely Important". The consensus was defined as a standard deviation $\leq 1.0$. Statements not achieving consensus were administered again in a third round between February 17th and March 9th, 2023; experts were shown the statement, their answer in the second round, and the mean of the score assigned by all raters. Experts were able to confirm their previous rating or revise it on the same scale.

## Data analysis

Round administration and data analysis were performed using Stat59 (Stat59 Services Ltd, Edmonton, AB, Canada). Categorical data were described through their distribution frequency.

For each statement, answers were analyzed through mean, standard deviation, and frequency distribution. Statements not reaching consensus (SD > 1) after the third round were classified as "consensus not reached".

## Results

Out of 95 authors contacted, 9 (9.5%) answered positively and were included after verifying their eligibility. Six eligible experts were recruited from the investigators' research network or referred by other contacted authors. Further 10 experts were recruited at the EUSEM 2022 congress, totaling 25 who consented to participate.

The first round was launched on October 22$^{nd}$, 2022, and lasted one month, collecting 79 statements from 15 experts (S1 Table). The remaining ten experts dropped out of the study without participating, reporting a lack of time or generic difficulties. The 15 experts completed all the rounds. Their age ranged between 38 and 75 years; most of them came from Australia (n = 4), Belgium (n = 3), and Italy (n = 2) and mainly were combined experts in emergency medicine and disaster medicine (n = 10) (S2 Table).

After content analysis, the 79 responses from the first round were condensed to 31 statements. The second round was launched on January 6$^{th}$, 2023, and lasted 40 days; 11 out of 31 statements reached immediate consensus. The remaining 20 statements were advanced to the third round between February 17$^{th}$ and March 9$^{th}$, 2023. At this final stage, 7 statements reached consensus, while 13 did not. A response rate of 100% was obtained.

Table 1 lists the 18 statements reaching consensus ordered by strength of agreement; those not reaching consensus are instead available in Table 2. Statements reaching consensus ranged between a rating of 6.1 ("Important") and 5.1 ("Somewhat Important").

## Discussion

In terms of quality, the study achieved current literature recommendations on sample size including more than 12 experts, which worked in different countries worldwide as clinicians or researchers. Most had combined emergency and disaster medicine expertise, but also public health. The heterogeneity of backgrounds and countries suggests that multiple perspectives on the same problem could have been included.

Most experts came from high-income countries, mirroring the lack of published research on the health impact of heatwaves from developing countries [3]. Probably due to their background–overlapping with humanitarian medicine–the experts agreed that disparity between different income-level countries is a factor causing emergency care overload (#14). Given the lack of literature in the field from low-income countries, and therefore no demonstrated association between the country's income level and the degree of emergency care demand during heatwaves, this statement seems not to identify an issue to be addressed to increase the resilience of emergency medical care systems. On the other hand, statement #14 offers the opportunity to reflect on the vulnerabilities of resource-poor countries in facing heatwaves [11]. Also, it raises the attention on future threats deriving from climate change, such as massive migrations due to regions becoming uninhabitable, that should be included in hazard and vulnerability analyses (HVAs) as phenomena potentially destabilizing emergency care systems in middle- and high-income countries.

**Table 1. Statements that attained consensus.**

| # | Statement | Round | Mean | SD | Patient Processing Section (Intervention Target) |
|---|-----------|-------|------|-----|--------------------------------------------------|
| 1 | Availability of air conditioning at home or access to cooled spaces can reduce heat-related illnesses. | 2 | 6.3 | 0.9 | Input (Population) |
| 2 | Information campaigns about Heatwaves can help increase community awareness and educate on how to deal with extreme temperatures. | 3 | 6.2 | 1.0 | Input (Population) |
| 3 | During heatwaves, there is an increased number of presentations to the emergency department to be managed. | 2 | 6.1 | 0.9 | Input |
| 4 | Emergency planning and surge capacity development are important to prevent emergency medical care overload during heatwaves. | 3 | 6.0 | 1.0 | Throughput |
| 5 | Health systems already stretched by other coexisting medical conditions (e.g., COVID-19) could be more fragile if affected by heatwaves. | 2 | 5.9 | 0.9 | Input (Stakeholders) |
| 6 | Awareness of stakeholders and policymakers is important to mitigate heatwaves effects on emergency medical care overload. | 2 | 5.9 | 0.6 | Input (Stakeholders) |
| 7 | Training of healthcare personnel - working in primary care, emergency medical service, and emergency department - could improve heat-related illness diagnosis and treatment. | 2 | 5.9 | 1.0 | Throughput |
| 8 | Avoid planning of public events/mass gatherings during forecasted heatwaves. | 2 | 5.8 | 1.0 | Input (Stakeholders) |
| 9 | Primary care is responsible for identifying at-risk population strata and enacting interventions to prevent heat-related illnesses. | 3 | 5.8 | 1.0 | Input (Primary Care) |
| 10 | Prevention of chronic medical conditions exacerbations can reduce the burden on emergency medical care during heatwaves. | 2 | 5.7 | 0.9 | Input (Primary Care) |
| 11 | Interventions dedicated to emergency medical services and emergency departments into Heat Action Plans can improve the overall heatwaves response. | 2 | 5.7 | 0.7 | Throughput |
| 12 | Implementing telemedicine during heatwaves for fragile individuals in the community could improve patients' management through primary care. | 2 | 5.6 | 0.7 | Input (Primary Care) |
| 13 | During heatwaves there is an increased demand for emergency medical services (emergency calls, ambulance dispatches) to be managed. | 3 | 5.6 | 1.0 | Input |
| 14 | Investments are needed to minimize disparities between high- and middle/low-income countries regarding response to heatwaves. | 3 | 5.5 | 1.0 | - |
| 15 | The deployment of additional resources during heatwaves should be proportional to the prevalence of vulnerable/frail individuals found in the community. | 2 | 5.3 | 1.0 | Throughput |
| 16 | The reduction of hospital bed capacity during summer can worsen emergency medical care overload during heatwaves. | 3 | 5.3 | 0.9 | Output |
| 17 | Planning additional equipment in emergency medical service and emergency department is important when heatwaves are forecasted. | 3 | 5.2 | 1.0 | Throughput |
| 18 | Planning additional staff in emergency medical service and emergency department is important when heatwaves are forecasted. | 2 | 5.1 | 1.0 | Throughput |

The overload experienced by emergency care during heatwaves mirrors the broader term of ED overcrowding, which affects acute care also in normal times. Several authors tried to analyze and find solutions to ED overcrowding by adopting different conceptual models. The most diffused model of input-throughput-output proposed by Asplin et al. applies operations management concepts to patient flow [12], and can be used to classify the findings of the present Delphi study.

## Input section

Most of the statements achieving consensus and the only three statements attaining a mean score > 6 (Important) seemed to deal specifically with the input part of patient processing, probably in the search for solutions to prevent the surge rather than trying to expand the capacity of an emergency system–almost exhausted already in non-crisis times. In fact, the Delphi experts reiterated the importance of the surge of patients requesting care during extreme heat periods in causing overload (input)–in terms of increased demand to EMSs (#13)

**Table 2. Statements which did not attain consensus.**

| Round | Statement |
|---|---|
| 3 | Extended availability of primary care physicians during heatwaves could divert minor issues from the emergency medical care system. |
| 3 | Deploy additional primary care personnel to take care of frail persons and high-complexity cases (multiple diseases, high number of medications, social reasons), to avoid ambulance overuse and transport to the ED. |
| 3 | Systematic detection of patients' body temperature in primary care and emergency medical care during heatwaves could help diagnose heat-related illnesses. |
| 3 | Heatwave warning communications can educate the population on the appropriate use of primary care / emergency medical service / emergency department during heatwaves. |
| 3 | Heatwaves period coinciding with annual leaves/vacations of healthcare personnel can affect staff retrieval during heat-related surges. |
| 3 | Planning the repurposing of emergency medical service and emergency department is important when heatwaves are forecasted. |
| 3 | Digitalization and communication improvements between emergency medical care and emergency department can reduce emergency medical care overload during heatwaves. |
| 3 | Training about heatwaves specific features delivered to incident command systems' personnel can reduce emergency medical care overload during heatwaves. |
| 3 | Deploy dedicated personnel to deal with high-complexity patients (e.g., with comorbidities, high number of medications, limited family/social support) presenting to the Emergency Department during heatwaves to relieve the department's overload. |
| 3 | Emergency medical service personnel are at risk of suffering from acute heat-related illness when dispatched to the prehospital arena. |
| 3 | Daily variations in heat-related illnesses helps in resource adaptation. |
| 3 | Activation of dedicated beds to treat heat stroke patients in the Emergency Department during heatwaves is important. |
| 3 | Implementation of syndromic surveillance systems focused on diseases caused/exacerbated by heatwaves can be used to adapt resources. |

and presentations to the EDs (#3)–as already demonstrated by several authors [6]. Such a tendency of the panel could be interpreted as that the excessive demand and burden on EDs and EMSs could be tackled by enacting public health initiatives or preventive healthcare measures delivered to the most vulnerable before acute care. Within the statements attaining consensus, experts therefore identified three main targets: stakeholders / policymakers, the population itself, and primary care.

**The role of stakeholders and policymakers.** The experts confirmed the importance of improving the awareness of stakeholders and policymakers (#6) since they have the responsibility of creating heat-health action plans (HHAPs) or revising those already existing to prepare all the sectors of healthcare [13]. Of note, experts highlighted the need to reshape healthcare systems already stretched by other coexisting medical conditions (e.g., COVID-19) as being more fragile if affected by heatwaves (#5). In light of the future recurrence of heatwaves and their possible overlap with other concurrent disasters amplifying detrimental effects [14], HHAPs should be flexible, dynamic, and based on frequent, periodic HVAs.

An interesting statement that obtained consensus targeted public events and mass gatherings during forecasted heatwaves (#8) as a potential source of overload for emergency care. Particular attention should be paid to which healthcare resources prediction model is used since only some tools include variables such as temperature and humidity [15]. Also, given the potential short notice of weather forecasts, local authorities should consider postponing events coinciding with heatwaves to avoid multiple casualties and an unpredicted strain on the emergency care system.

**The role of the population.** As HHAPs contain several interventions triggered by warning systems to preserve population health during heatwaves [16,17], the experts agreed on

some as potentially important to relieve the burden on emergency medical care. Heatwaves-awareness campaigns and alerts are a critical element in documents providing guidance on HHAPs [17,18]. Standardized messages can be effectively delivered through different media but campaigns have to consider factors hampering information access, such as higher age, unemployment, or low education [19]. The experts, in fact, considered information campaigns to raise community awareness and educate the population as important (#2). Despite limited evidence, awareness campaigns and structured heat warnings seem to contribute to reducing mortality [20] and could help in preventing heat illness or the worsening of chronic diseases, which represent the majority of emergent presentations during heatwaves. On the other hand, the experts did not agree on the potential of warning communications in educating the population on the appropriate use of primary care, EMSs, and EDs during heatwaves (Table 2), probably because this aspect is already difficult to establish in non-heatwaves times [21] and has still to be studied. Anyway, it could be reasonable to update HHAPs with statements raising the awareness of the population on the burden posed by heatwaves on the healthcare system and promoting both prevention and a more appropriate use of healthcare.

Availability of air conditioning at home or access to cooled spaces can reduce heat-related illnesses, especially among vulnerable populations. The experts agreed that air conditioning could translate into a reduced burden on emergency care (#1), probably by preventing heat illness development. Unfortunately, a reduced access to air conditioning has been associated with lower socio-economic status and poorer quality of housing [22]. Electric fans are cheaper, but current literature is insufficient to recommend such devices as a valid method to reduce heat during heatwaves [23]. Cooling centers can be another alternative, but prejudices hampering their attendance have been reported [5]. Therefore, access to air conditioning could be considered a factor improving population health during heatwaves, but not the main solution.

**The role of primary care.** Discrepancies in risk perception and self-protecting behaviors during heatwaves have been noted in the literature, especially in the elderly and subjects with comorbidities being not aware of their frailty [4,5]. Also, chronic diseases undoubtedly contribute to ED overload during ordinary times [9] and constitute a significant part of patients seeking care during heatwaves [4,8]. Experts agreed that primary care should identify these vulnerable strata and enact interventions to prevent heat-related illnesses (#9) and that preventing chronic medical conditions exacerbations during heatwaves could reduce the burden on emergency medical care (#10). These two statements are in line with current recommendations, suggesting that the at-risk population should be identified before heatwaves occur [24], which could make warning systems more efficient from the earliest phases. Interestingly, experts also deemed telemedicine implementation important to improve fragile individuals' management during heatwaves by primary care in the community (#12). Such innovative technology could not be available in every setting, but improving at least telephone contact through a dedicated heat-help line could allow the management of mild cases at home and deliver tailored counseling, especially for the elderly. Experts did not agree on the usefulness of extending the availability of primary care physicians or deploying more primary care personnel (Table 2), in contrast with current literature, suggesting that ED visits could be reduced by addressing barriers inherent to a timely access to primary care [25]. The experts probably interpreted these statements as two isolated and one-sided factors/interventions during the Delphi process, thus disagreeing. Wider integrated interventions, with population and stakeholders' engagement, could instead be more effective in improving access to primary care. In general, the contribution of primary care during heatwaves can be fundamental [26]. Its role in preserving emergency medical care from overload and failure should be therefore addressed explicitly in HHAPs and measured by future studies.

## Throughput section

Knowing from previous analyses that HHAPs lack uniformity [27], it can be expected that not every plan specifically entails emergency medical care-tailored interventions to improve patient flow, which instead were deemed important by the experts to improve ED and EMS response to heatwaves (#11). By following disaster medicine principles, HHAPs should contain measures to develop surge capacity (#4) in different ways so to improve patient throughput [9]. Experts agreed also on planning additional equipment (#17) and staff (#18) in both EMSs and EDs when heatwaves are forecasted, consistently with the "staff, stuff, and structure" surge science framework [28], and proportionally to the prevalence of vulnerable/frail individuals in that community (#15). Of note, experts did not agree on deploying dedicated personnel to care for high-complexity, comorbid patients in the ED or to create "heat stroke" beds in the ED (Table 2), probably because they were more concerned by the volumes of all-comers (more predictable) rather than by specific cases (less predictable). Along with more resources, adequate training across primary and emergency medical care has been identified as important by the panel (#7). Dedicated training proved to increase heat illness recognition and treatment in the ED [29], but could easily be exported to other healthcare contexts. Experts did not agree on systematically measuring body temperature (Table 2), probably because skin temperature is not accurate in heat illness detection, and core body temperature (rectal), despite being the most accurate [30], is unfeasible as a screening tool. Therefore, the effects of mandatory training for primary and emergency care practitioners with a refresher provided once a year (e.g., before the hot season), or adjunctive triage questions triggered by HHAPs activations could be studied.

## Output section

Regarding output in patient processing, the reduction of hospital bed capacity during summer has been identified as potentially worsening emergency medical care overload during heatwaves (#16). Seasonal hospital adaptations meet fluctuations in diseases throughout the year, usually by expanding bed capacity and reducing elective procedures during winter to admit respiratory illnesses [31]. Since one of the most critical factors causing ED overcrowding is hospital boarding [9], healthcare systems with a reduced bed capacity during the warm season and coinciding with heatwaves could register significant delays in patient output from the ED. Hospital boarding also delays throughput since personnel is diverted to care for patients while waiting to be admitted to the wards, and stretchers are unavailable to receive new patients from EMS–automatically reducing ambulance availability. A reasonable approach could therefore consider dynamically adapting bed capacity when HHAPs are activated, similar to hospital repurposing during the COVID-19 pandemic, mitigating the boarding bottleneck.

## Final recommendations

Overall, a list of recommendations can be summarized from the consensus reached among experts:

- HHAPs should be flexible, dynamic, and based on frequent, periodic HVAs to meet climate change consequences

- HHAPs should contain interventions raising the awareness of the population on heatwave-related risks and preventing heat illnesses towards a more appropriate use of healthcare system's resources

- Develop telemedicine dedicated to the frail strata of the population (e.g., a telephone helpline for counseling or at-home management)

- HHAPs should contain interventions dedicated to primary care, which has a fundamental role in preserving emergency medical care from overload

- Local authorities should consider postponing events coinciding with heatwaves to avoid multiple casualties and an unpredicted strain on the emergency care system.

- Training on heat illnesses for primary and emergency care practitioners before every hot season

- Adapt hospital bed capacity when HHAPs are activated

## Limitations

The results derived from this Delphi study are specific to the panel of experts who participated and could be invalid for other contexts. For example, there was no expert from Southern America or Africa, and 14 out of 15 were from high-income countries. In line with both UN Sustainable Development Goals #1, "No Poverty", and #13, "Climate Action"–low- and middle-income countries should be given the possibility to improve research and find more specific strategies against heatwaves, so a dedicated Delphi process could help filling this gap.

## Conclusions

Heatwaves, as a consequence of climate change, are affecting population health and healthcare systems worldwide. In particular, emergency medical care suffers from an overload of patients to be processed during heatwaves, but studies testing solutions to mitigate such burden are lacking. Since Delphi studies are performed to gather information in fields with poor or conflicting knowledge, this methodology was used to identify factors causing emergency care overload during heatwaves as a potential target of future interventions. The Delphi process provided 18 statements, mostly focused on the input section of patient processing, identifying stakeholders, the population, and primary care as targets of future discussion and interventions such as HHAPs implementation, training of healthcare personnel, or education and awareness campaigns. Additional dedicated resources and bed capacity were deemed important as per throughput and output sections, respectively.

The next steps in relieving emergency care strain during heatwaves should entail the evaluation of these statements by national and international scientific societies to be implemented and tested in EDs and EMSs.

## Supporting information

**S1 Table. First round: 79 statements proposed by the 15 included experts to the question "*What are the factors causing emergency medical care overload during heatwaves*?".** (DOCX)

**S2 Table. Included experts' characteristics.** (DOCX)

## Acknowledgments

The present manuscript results from a study conducted in the framework of the International Ph.D. in Global Health, Humanitarian Aid, and Disaster Medicine jointly organized by Università del Piemonte Orientale (UPO) and Vrije Universiteit Brussel (VUB).

## Author Contributions

**Conceptualization:** Matteo Paganini, Hamdi Lamine, Francesco Barone-Adesi.

**Data curation:** Matteo Paganini, Hamdi Lamine.

**Formal analysis:** Matteo Paganini.

**Investigation:** Matteo Paganini.

**Methodology:** Matteo Paganini, Francesco Barone-Adesi.

**Project administration:** Luca Ragazzoni.

**Resources:** Matteo Paganini.

**Software:** Matteo Paganini.

**Supervision:** Hamdi Lamine, Francesco Della Corte, Ives Hubloue, Luca Ragazzoni, Francesco Barone-Adesi.

**Validation:** Luca Ragazzoni.

**Visualization:** Luca Ragazzoni, Francesco Barone-Adesi.

**Writing – original draft:** Matteo Paganini.

**Writing – review & editing:** Matteo Paganini, Hamdi Lamine, Francesco Della Corte, Ives Hubloue, Luca Ragazzoni, Francesco Barone-Adesi.

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
