## [Decision Letter · Decision Letter 0]

18 Oct 2023

PONE-D-23-26567Factors Causing Emergency Medical Care Overload During Heatwaves: A Delphi StudyPLOS ONE

Dear Dr. Paganini,

Thank you for submitting your manuscript to PLOS ONE. After careful consideration, we feel that it has merit but does not fully meet PLOS ONE’s publication criteria as it currently stands. Therefore, we invite you to submit a revised version of the manuscript that addresses the points raised during the review process.

We look forward to receiving your revised manuscript.

Kind regards,

Yong-Hong Kuo

Academic Editor

PLOS ONE

Journal Requirements:

Additional Editor Comments:

Two referees have reviewed the paper and, in general, believe that the paper has the potential to be published. They have kindly provided some constructive comments for the authors to address in their revision. Please address the concerns seriously for consideration for possible publication.

Reviewers' comments:

Reviewer's Responses to Questions

**Comments to the Author**

1. Is the manuscript technically sound, and do the data support the conclusions?

Reviewer #1: Yes

Reviewer #2: Yes

2. Has the statistical analysis been performed appropriately and rigorously? 

Reviewer #1: Yes

Reviewer #2: Yes

3. Have the authors made all data underlying the findings in their manuscript fully available?

Reviewer #1: Yes

Reviewer #2: Yes

4. Is the manuscript presented in an intelligible fashion and written in standard English?

Reviewer #1: Yes

Reviewer #2: Yes

5. Review Comments to the Author

Reviewer #1: The authors conducted a Delphi study to derive the causes of emergency medical care overloading during heatwaves. Based on the analysis of the consensus 18 statements, the main factors are around patient

processing, identifying stakeholders, the population, primary care, as well as dedicated resources and bed capacity. My recommendation is major revision given my following comments:

(1) As a research study, evaluation/validation of the results is an indispensable section. However, the authors claim that the evaluation is their next step. From my point of view, evaluation/validation should be added to make this study solid and compact.

(2) The Conclusion section is weak since it just repeats the contents in the abstract. Please revise it to present a more comprehensive and stronger summary.

(3) The Discussion section is comprehensive, but the presentations need improvements. From the current version, it is hard to efficiently grasp the core findings and insights. This section should be more systematic.

(4) In Table 1, the mean values seem to reflect the significance of statements. Are there any insights from the mean metric?

(5) Based on the mostly focused aspects authors highlighted, Table 1 could be better if a column of statements' corresponding aspects (i.e. identifying stakeholders, the population, primary care, etc.) is added. In this way, each statement can be directed to the associated aspect.

In a word, I suggest a major revision. More efforts are needed to improve this study.

Reviewer #2: It would be valuable to frame a limitation section that explicitly addresses the geographical bias, given that a majority of the experts participating in this Delphi study are from high-income countries.

6. PLOS authors have the option to publish the peer review history of their article (what does this mean?). If published, this will include your full peer review and any attached files.

Reviewer #1: No

Reviewer #2: No

---

## [Author Response · Author response to Decision Letter 0]

27 Oct 2023

Response to reviewers provided as a separate file

---

## [Decision Letter · Decision Letter 1]

16 Nov 2023

Factors Causing Emergency Medical Care Overload During Heatwaves: A Delphi Study

PONE-D-23-26567R1

Dear Dr. Paganini,

We’re pleased to inform you that your manuscript has been judged scientifically suitable for publication and will be formally accepted for publication once it meets all outstanding technical requirements.

Kind regards,

Yong-Hong Kuo

Academic Editor

PLOS ONE

Additional Editor Comments (optional):

The concerns have been addressed. I recommend Accept.

Reviewers' comments:

Reviewer's Responses to Questions

**Comments to the Author**

1. If the authors have adequately addressed your comments raised in a previous round of review and you feel that this manuscript is now acceptable for publication, you may indicate that here to bypass the “Comments to the Author” section, enter your conflict of interest statement in the “Confidential to Editor” section, and submit your "Accept" recommendation.

Reviewer #1: All comments have been addressed

2. Is the manuscript technically sound, and do the data support the conclusions?

Reviewer #1: Yes

3. Has the statistical analysis been performed appropriately and rigorously? 

Reviewer #1: Yes

4. Have the authors made all data underlying the findings in their manuscript fully available?

Reviewer #1: Yes

5. Is the manuscript presented in an intelligible fashion and written in standard English?

Reviewer #1: Yes

6. Review Comments to the Author

Reviewer #1: After the revision and the authors' explanations, all my comments have been addressed. The structure of this study is clearer and much better. Therefore, my recommendation is Accept.

7. PLOS authors have the option to publish the peer review history of their article (what does this mean?). If published, this will include your full peer review and any attached files.

Reviewer #1: No

---

## [Editor Report · Acceptance letter]

11 Dec 2023

PONE-D-23-26567R1 

Factors causing emergency medical care overload during heatwaves: A Delphi Study 

Dear Dr. Paganini:

I'm pleased to inform you that your manuscript has been deemed suitable for publication in PLOS ONE. Congratulations! Your manuscript is now with our production department. 

Kind regards, 

on behalf of

Dr. Yong-Hong Kuo 

Academic Editor

PLOS ONE